# MAML: Towards a Faster Web in Developing Regions

## Abstract

The web experience in developing regions remains subpar, primarily due to the growing complexity of modern webpages and insufficient optimization by content providers. Users in these regions typically rely on low-end devices and limited bandwidth, which results in a poor user experience as they download and parse webpages bloated with excessive third-party CSS and JavaScript (JS). To address these challenges, we introduce the Mobile Application Markup Language (MAML), a flat layout-based web specification language that reduces computational and data transmission demands, while replacing the excessive bloat from JS with a new scripting language centered on essential (and popular) web functionalities. Last but not least, MAML is backward compatible as it can be transpiled to minimal HTML/JavaScript/CSS and thus work with legacy browsers. We benchmark MAML in terms of page load times and sizes, using a *translator* which can automatically port any webpage to MAML. When compared to the popular Google AMP, across 100 testing webpages, MAML offers webpage speedups by tens of seconds under challenging network conditions thanks to its significant size reductions. Next, we run a competition involving 25 university students porting 50 of the above webpages to MAML using a web-based *editor* we developed. This experiment verifies that, with little developer effort, MAML is quite effective in maintaining the visual and functional correctness of the originating webpages.

## 1 Introduction

The modern web has undergone a profound transformation over the past decade, largely driven by the proliferation of client-side interactions and the widespread adoption of development frameworks such as React [48] and Angular [33]. While these frameworks were originally intended to handle large-scale, feature-rich applications, they have increasingly become default choices even for smaller, less complex projects. As a result, the very nature of web complexity has shifted, as what used to be a basic interface element is now part of a large library, importing excessive stylesheets and scripts. This web "bloat" has been further exacerbated by the widespread use of third-party scripts from content delivery networks (CDNs), analytics services, and other external resources. Butkiewicz et al. [10] showed that, on average, modern web pages rely on at least 5 non-origin sources, contributing to more than 35% of the total bytes downloaded.

In addition to heavier resource demands, modern web applications often present intricate Document Object Model (DOM) trees, which require browsers to perform intensive computations to find and update individual elements—a process collectively known as "reflows and repaints". Although best practices to avoid these complex computations are proposed [22, 45, 47], the fundamental issue of web complexity persists. This growing complexity is not merely

a technical concern—it poses a direct challenge to users in developing regions, where low-end smartphones and limited internet infrastructure dominate. With the need for efficient bandwidth use and resource optimization, these users are often excluded from fully benefiting from modern web experiences, which tend to cater to high-end devices and fast, stable internet connections.

Recognizing these challenges, developers and organizations are increasingly advocating for more streamlined web development strategies. Notable initiatives such as Google AMP [3], Facebook Instant Articles [4], and SpeedReader [26] change a webpage's layout to optimize the display and functionality of its components. Additionally, research works [11, 42, 50] focus on optimizing the DOM tree to reduce its depth and complexity, thereby enhancing overall page performance. However, these methods still heavily depend on resource management and JavaScript (JS) execution, which can adversely affect performance, especially on devices with limited processing power.

This paper proposes MAML (Mobile Application Markup Language), a new mobile web specification language designed to speed up webpages in developing regions. MAML is based on three core founding principles. First, it adopts a "flat" DOM structure, using absolute positioning to place elements relative to the viewport, thereby reducing computational complexity and eliminating dependencies on surrounding elements. Second, it introduces a new scripting language to eliminate excessive bloat from JS, focusing only on essential functionalities to reduce complexity and avoid unnecessary code. Third, it supports transpilation to minimal HTML/JS/CSS, thus making it backward compatible with today's web ecosystem.

We developed a web-based MAML *editor* to assist developers in creating webpages that conform to MAML specifications. Additionally, we developed a *translator* that automates roughly 65% – empirically estimated in our study – of the manual tasks required to convert existing webpages into MAML. The remaining 35% mostly relates to page interactions triggered via human inputs, which can be easily implemented via our visual editor. MAML translator further supports transpiling MAML code back to minimal HTML/JS/CSS, so that it can be served and rendered by today's browsers. MAML translator can thus be adopted by CDN providers or acceleration proxies like [28] and [5] to improve their users experience by serving MAMLed webpages.

We use the MAML *translator* to convert existing pages to MAML and benchmark MAML on user QoE and bandwidth savings compared to Google AMP, and visual similarity of MAML webpages compared to the original webpages. We find that MAML pages load significantly faster across all timing metrics and generates a median data saving of 1 MB compared to AMP and 2.4 MB compared to the original. In addition, MAML outperforms AMP by up to 30% in terms of visual similarity. Next, we conduct a competition among computer science students to customize MAML pages using our *editor* verifying that webpages converted to MAML format outperform on all timing metrics and consume less data, which is particularly beneficial in bandwidth-constrained regions.

*WWW '25, April 28–May 02, 2025, Sydney, Australia*
2018. ACM ISBN 978-1-4503-XXXX-X/18/06
https://doi.org/XXXXXXX.XXXXXXX

## 2 Background and Related Work

**Challenging Network Conditions.** Poor web performance for users in developing regions can be triggered due to network-induced delays. Zaki et al. [53] showed that developing regions like Ghana suffer from long page load times due to HTTP redirects, DNS lookups, and TLS/SSL connection setups. Koradia et al. [38] demonstrated that cellular data connectivity in India suffers from significantly high latencies of up to 1,200 ms. Chen et al. [18] showed that, in bandwidth-constrained environments, TCP flows experience severe unfairness, high loss rates, and flow silences due to repetitive timeouts, resulting in poor web performance. Other studies have pointed out traffic engineering and lack of infrastructure as reasons for high delays [15, 25, 27, 32]. Feamster et al. [21] have also demonstrated that in South Africa, despite the presence of web caches, users experience high latencies because the connectivity within the country is still limited.

**Web Complexity.** The complexity of webpages is another issue behind the high page load times in developing regions. Indeed, modern webpages consist of a large number of web elements hosted across several domains. Butkiewicz et al. [10] have shown that more than 60% of webpages request data from at least 5 different non-origin sources, contributing to more than 35% of the overall page size. Furthermore, a modern browser must fetch and render several objects, including HTML, JS, CSS, and images, forming a complex object dependency graph [11, 42, 49]. Every modification to the DOM—whether through adding or removing elements, changing attributes, altering classes, or executing animations—triggers the browser to recalculate styles and adjust part or all of the layout, collectively referred to as "reflows and repaints." This requires the browser to match selectors against the elements in the DOM to determine which CSS rules apply. This resource-intensive operation can significantly slow down a webpage load, especially on low-end devices which are common in developing regions.

**Absolute Position-Based Web Development.** An approach that avoids the constraints of the DOM structure is "absolute positioning" [40]. Absolute position-based web development is generally viewed with caution and used by developers only when required. It leverages CSS's absolute positioning to place elements with precise control, allowing designers to position elements relative to the viewport. This approach can create complex layouts more easily, but it also introduces challenges related to responsiveness. Elements positioned absolutely are removed from the normal document flow, meaning they do not adapt to changes in the surrounding components or viewport size. As a result, reliance on absolute positioning can complicate the standardization of layout processes. Nevertheless, absolute positioning enables the creation of a "flat" DOM layout, modularizing the components and enabling a more efficient means of searching and updating elements on the viewport.

**Optimizations.** Several techniques have been proposed to optimize web browsing over challenging networks, including network-level optimizations, caching techniques, and content distribution mechanisms [16, 17, 19, 20, 35, 46]. Recent works [11, 42, 50] have focused on the complexity of webpages and suggested different approaches to address them. Many solutions have been proposed to optimize

the usage of JS in modern webpages [13, 14, 39], especially focusing on identifying and blocking unused and non-essential JS code.

From a product perspective, Google AMP [3] rewrites webpages with new HTML tags and elements; Facebook Instant Articles [4] enables publishers to create fast and interactive articles; and Opera mobile browser [5] compresses pages by about 90% on Opera servers before they are transferred to the client's device, rendering them faster by 2-3 times. Google Web Light [28], though discontinued, served a similar goal by transforming heavy web pages into lightweight versions to enhance performance on slower networks. SpeedReader [26], unlike traditional reader modes, integrates directly into the rendering pipeline to improve both performance and privacy by stripping unnecessary elements before rendering.

MAML fundamentally differs from the above approaches in that it pre-compiles pages to simplify their HTML representation, eliminating recursive handling of objects and simplifying the DOM to a flatter layout with absolute positioning while maintaining the original functional equivalence. We believe that reliance on HTML, JS, and CSS is the underlying problem, and none of the above solutions tackles this fundamental issue.

## 3 Motivation and Challenges

It is known that webpage complexity has significantly increased in the past few years [44]. However, the impact of this on users in developing regions with low-end mobile devices and limited network connectivity has not been given sufficient attention. To assess today's webpage complexity issues, we identified 100,000 websites of developing regions from the Chrome User Experience Report (CrUX) dataset [23]. We gathered data for ten developing countries (10,000 webpages per country) classified as the ten most populous "developed" nations by the IMF [34]. To classify websites by country, we used the domain's Whois information, and the country's top-level domain (e.g., .pk for Pakistan).

We measured these websites using Google Lighthouse [1], an open-source tool for audits across multiple dimensions like webpage performance and composition. Specifically, we gathered the following metrics: speed index, number of DOM elements, maximum number of DOM depth between all children inside the <html> tag, number of stylesheet requests, number of script requests, total CSS size, and total JS size. A low-end mobile device was emulated to access the sites using Lighthouse [1], with network conditions set to 3G Fast (1.6 Mbps downlink/768 Kbps uplink with 150ms RTT). This network condition is the "average" network configuration that we used in our benchmark evaluation (see Section 6).

Figure 1 shows a radar-plot of the aforementioned metrics, categorizing each metric according to Lighthouse's color coding scheme of *speed indexes* [30], a metric measuring how quickly a webpage is visually complete above-the-fold. A value in the range of 0 to 3.4 sec is classified as green, indicating optimal performance. A value between 3.4 and 5.8 sec is classified as orange, suggesting moderate performance with potential areas for improvement. A value greater than 5.8 sec is categorized as red, indicating poor performance that requires immediate attention and optimization.

Our evaluation shows that 47.3% of webpages fell in the red region (*i.e.,* speed index > 5.8 sec), 28% in the orange, and 24.6% in the green region. The median number of DOM elements in the

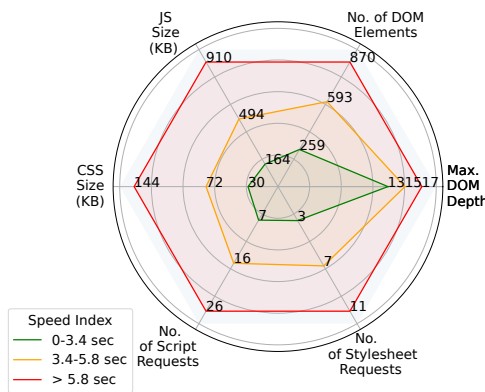

**Figure 1: SI versus median complexity metrics as measured via Lighthouse for 100k developing regions websites.**

red region is 910, with a median of 26 script requests. The median maximum DOM depth is 17 and is consistent across other speed index categories. Additionally, the median size of JavaScript is 6.3 times larger than that of CSS.

These findings underscore the critical nature of web performance issues: as the webpage complexity and heavy reliance on JS frameworks increases, greater memory allocation and computational resources are required for DOM manipulations and CSS recalculations, thereby elongating the rendering cycle and degrading user-perceived performance. In turn, the speed index deteriorates, leading to a slower loading time and a detrimental user experience. On low-end mobile devices with limited resources, which are dominant in developing regions, the impact of this becomes even more pronounced. These devices, constrained by slower processors and less RAM, struggle with the increased computational load, leading to longer page load times and more frequent browser crashes or unresponsive webpages. In essence, this correlation indicates that work is rather necessary to fundamentally redefine how webpages are created, ensuring efficiency across all device types in all regions.

## 4 The MAML Language

This section introduces MAML, an innovative web specification language designed to enable the creation of fast-loading webpages while maintaining simple interactive features.

### 4.1 Design Principles

MAML follows several design principles rooted in the desire to reduce complexity and optimize performance. At its core, MAML seeks to create a more intuitive and straightforward way to build web applications, achieving two key objectives: 1) less development overhead, and 2) faster load times in areas with slow internet, where saving bandwidth is crucial for lowering costs. We plan to achieve this through the following principles:

**Flat DOM**: Hierarchical DOM trees, as present in today's web, involve complex layout recalculations that can strain low-powered devices. To solve this problem, MAML adopts a flat DOM approach, which minimizes the depth of the DOM tree and reduces computational complexity. Elements are placed in absolute positions (see Section 2) relative to the viewport, and their layout and style

configurations do not depend on surrounding elements. This simplification is particularly beneficial for devices with limited processing power as it eliminates the need for complex layout recalculations. To solve the challenge of responsiveness to dynamic screen sizes, we use a proportional scaling technique to reposition and rescale elements appropriately across different viewports.

**Bloat Avoidance**: Modern webpages often include large amounts of unnecessary code, such as unused CSS or complex JS libraries. MAML avoids this by limiting the range of supported attributes and completely cutting off JS, thereby avoiding useless page bloat and complexities by only focusing on essential functionalities.

**Backward Compatibility**: MAML aims to be backward compatible with today's Web and thus run on legacy browsers. Accordingly, we require that MAML can be *transpiled*[1] to regular, but minimal, HTML/JavaScript/CSS, and can thus be easily adopted by developers integrating it directly into their development workflows.

### 4.2 Flat DOM

**Data Structure:** MAML introduces a new format for writing webpages based on a *flat* DOM, where each element retains necessary information and attributes related to itself in a self-contained dictionary representation. Each element is a hash map containing key-value pairs, where the key is the element's property and the value is the property's assigned value. The use of a hash map data structure ensures that accessing the value of a property has a time complexity of $O(1)$. The resulting MAML file (or the MAML version of a webpage) is a collection of MAML data structures separated by a newline character (\n) and has an extension of *.maml*.

In the example below, the MAML page has a single element of type "image", with attributes related to the position of where that image should be displayed on the webpage viewport (i.e., the x and y coordinates of the upper left corner pixel of the image). In addition, MAML also specifies the z coordinate to establish element order in terms of depth, whereas the size of the displayed element is represented by the width (w) and height (h). Finally, the image element also specifies the URL to the image source as well as the alternative text.

```
{"type":"img","w":268,"h":31,"x":336,"y":15,"z":1,
 "src":"https://example.com/img/abc.webp",
 "alt":"Alternate Text","fit":"fill"}
```

**Supported Elements.** MAML supports a wide variety of webpage components, including Text, Shape, Text Field, Button, Dropdown, Image, Carousel, Video, and Script. These elements cover everything from basic textual content and geometric shapes to interactive components like buttons and text fields, as well as media content such as images, carousels, and videos. Each element has several mandatory properties, as shown in Table 1, along with their descriptions. Based on the type of element, each MAML element includes its own specific set of additional properties, which are detailed in Table 2. MAML also incorporates MAMLScript (see Section 4.3), a scripting language tailored for dynamic content manipulation.

---

[1]Transpilation is the process of converting source code from one high-level programming language to another.

| Property | Description |
|---|---|
| type | type of element |
| x | x-position of element in pixels |
| y | y-position of element in pixels |
| z | z-position of element as integer |
| w | width of element in pixels |
| h | height of element in pixels |
| display | whether to make the item visible or not |

Table 1: Mandatory properties of MAML elements.

| Element | Available Properties |
|---|---|
| text | id, text, fontFamily, textAlign, fontSize, color, fontStyle, fontWeight, display |
| shape | id, backgroundColor, borderRadius, display |
| text-field | id, placeholder, backgroundColor, display |
| button | id, text, display |
| dropdown | id, options, display |
| image | id, src, objectFit, display |
| carousel | id, srcs, display |
| script | code |

Table 2: Additional properties of MAML elements.

**Dynamic Positioning.** The web today is inherently dynamic, with users accessing content on a wide range of screen sizes and resolutions. Designing for this variability requires adaptable layouts that maintain visual consistency across all devices.

MAML employs a proportional scaling approach to position and scale elements properly across different viewports. Each MAML file includes a `viewport_width` property at the top, which specifies the width of the original viewport on which the page was designed. Height is not required because scaling based on width alone maintains the aspect ratio, ensuring that elements do not become distorted. Additionally, responsive design principles prioritize width for layout adjustments, while height can vary based on the content within elements. If an element has original coordinates $(x, y)$ and dimensions $(w, h)$ on the original screen, they are scaled according to the new scaling factor, maintaining the relative proportions. The scaling factor ($S$) is calculated as:

$$S = \frac{W_{\text{original}}}{W_{\text{new}}}$$

where $W_{\text{original}}$ is the width of the original viewport, and $W_{\text{new}}$ is the width of the new viewport. The width and the $x$ position are updated as follows:

$$x' = x \times S$$

$$w' = w \times S$$

We do not need to update the y position and the height, as browsers support scrolling until the end of the page.

The value of `viewport_width` is retrieved via a simple JS property `window.innerWidth` injected into the HTML transpiled from MAML (see section 5.2). After the page fully loads, the JS code updates the inline CSS `width` and `left` properties of each element within the body tag by multiplying them by the scaling factor ($S$).

## 4.3 MAMLScript

One of MAML's design principles is to avoid webpage *bloat* via complete JS removal, similar to Brave's "block script" feature[2]. The side effect of this aggressive strategy is a lack of page interactivity which can severely hinder the user experience. In order to provide some page interactivity, we design a simpler scripting language, MAMLScript, that efficiently supports a limited but popular set of JS functionalities.

MAMLScript is included at the end of a MAML file within a *script* element that has a property named *code*. The value of this property contains the MAMLScript code. When the MAML file is parsed, this element gives information about the dynamic updates applied to various elements. This method simplifies the mapping of actions to specific page components, thus facilitating a more streamlined interactivity framework, as can be seen in the following example:

```
{
    "type":"script",
    "code":"MAMLScript is included here."
}
```

**Supported Functionalities:** MAMLScript only supports *popular* JS functions, which we identify by analyzing the most frequently used interactive features across popular websites that rely on JS. Our analysis is performed manually due to the lack of a tool capable of accurately assessing interactive elements, which often require contextual understanding and nuanced evaluation. We analyze 100 websites with most visitor traffic and page views according to Amazon's Alexa Web Ranking service [9]. While a sample size of 100 may seem limited, it serves as a valuable indicator of the primary features that are essential for user interaction across diverse platforms. Moreover, Alexa's ranking is globally inclusive, representing a broad spectrum of sites, including those from developing regions, such as China's qq.com and sohu.com, as well as Indonesia's okezone.com. With the potential for open-source contributions, there is an opportunity for the community to expand on this evaluation, allowing for a more comprehensive understanding of interactive functionalities over time.

Our analysis works as follows: 1) identify components that change automatically, 2) hover over various parts of the page, and 3) interact through clicks. When a noticeable visual change occurs, we verify whether it was JS related and, if so, include it in our dataset. After inspecting these 100 websites, we identify the most common interactive features, seven of which are currently supported by MAML: 1) drop-down menus, 2) infinite scroll (loading new content when reaching the bottom), 3) video players, 4) image carousels, 5) elements that appear after scrolling past a certain point, 6) countdown timers, and 7) notification pop-ups. Beyond those, we found other frequently used features like 8) scroll-triggered animations, 9) auto-animations, 10) theme-toggle buttons, and 11) video previews activated by hovering over thumbnails. MAMLScript currently does not support these features, but can be added in the future.

**Structure**: MAMLScript closely mirrors familiar programming constructs found in languages like JS, making it accessible to a broad

---

[2]https://community.brave.com/t/brave-shields-and-js-blocking/509941

| Listener | Usage |
|---|---|
| click | on("click", *element_id*) { [triggers...] } |
| change | on("change", *element_id*) { [triggers...] } |
| keydown | on("keydown", *element_id*, *key_name*) { [triggers...] } |
| reach | on("reach", *element_id*) { [triggers...] } |
| timer | on("timer", *seconds*) { [triggers...] } |

**Table 3: MAMLScript listeners and their usage.**

| Trigger | Usage |
|---|---|
| val | val(*element_id*); |
| show | show(*element_id*); |
| hide | hide(*element_id*); |
| swap | swap(*content*, *element_id*); |

**Table 4: MAMLScript triggers and their usage**

range of developers. This design choice not only reduces the learning curve but also enables developers to leverage their existing coding skills when working with MAML files.

```
on("click", "button1") {
    show("image2");
    hide("image1");
    swap(val("input3"), "text3");}
```

Above, we show a MAMLScript which configures a sequence of actions to be executed in response to a click event on "button1". Upon activation, the script first makes "image2" visible using the show("image2") trigger. Then, it hides "image1" from view with the hide("image1") trigger, ensuring that "image2" takes its place on the screen. Finally, the script swaps the text of "text3" to the value of the text input field "input3" using the swap(val("input3"), "text3") trigger.

**Listeners and Triggers:** MAMLScript uses an Event-Driven Programming (EDP) paradigm. Each functionality is determined via a listener followed by one or more triggers. In the above example, *on* is used to listen to an event "click" on element id "button1". Three functions (show, hide, and swap) are then triggered on "image2", "image1", and "text3" one by one. Additionally, MAMLScript supports nested triggers—triggers that return values can be used as a value for another trigger, as illustrated by the swap and val triggers used together in the same example. Table 3 and 4 show the available listeners and triggers along with their usage.

## 5 MAML Backward Compatibility

Backward compatibility is the final founding principle of MAML (see Section 4.1). This principle translates into support of transpiling from MAML into HTML/JS/CSS. MAML achieves this by using the MAML "translator" (see subsection 5.2). The translator sequentially converts each line of a MAML file into an equivalent HTML component including its attributes and inline CSS styles. For example, a MAML element defined as an image will be translated into an HTML  tag with appropriate attributes such as id, src, and inline styles. It finally converts MAMLScript to JS by sequentially parsing listeners and triggers and writing the equivalent JS code that handles events and interactions defined in the MAMLScript.

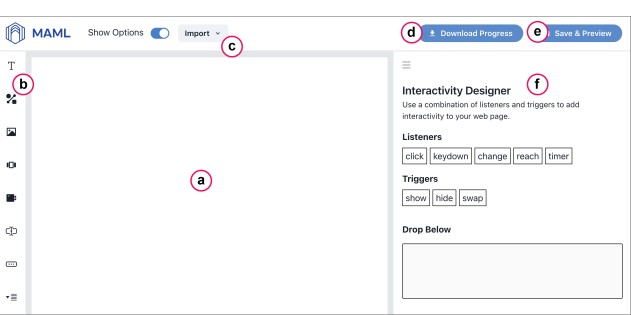

**Figure 2: MAML editor's user interface featuring a) canvas of size *1200×800px*; b) toolbar; c) import MAML file or existing url to translate; d) download a *.maml* file of the current design e) save & preview the resulting HTML version of the page; f) add interactivity using drag-and-drop listeners and triggers.**

To easily integrate MAML into a developer workflow, we have also developed a MAML *editor* (see subsection 5.2) based on a drag-and-drop interface mimicking other popular web editors, such as Wix [52], Elementor [24], Webflow [51], etc.

### 5.1 Editor

The MAML editor (see Figure 2) is a web application designed for both experienced and inexperienced web developers. The editor features a drag-and-drop interface to design the layout of a webpage and add interactivity to its elements.

**User Workflow.** Once users log in to the web-based MAML editor, they can: 1) create a MAML page from scratch; 2) import an existing *.maml* file from their local machine and customize it; or 3) import a URL that gets converted into the MAML format using the MAML "translator" (see subsection 5.2). To design the page layout, users can drag and drop elements from the sidebar onto the main canvas. Once an element is dropped, users see options to change the position, styles, and additional properties of the element.

To add interactivity to the elements, users can use the "interactivity designer". Users are required to drag and drop listeners and triggers to add event-driven behavior to the elements. For example, a user can set up a button to hide a specific image when clicked. The interactivity designer provides a visual interface for defining these interactions, making it easy for users to add dynamic functionality to their webpages. Once the page is complete, users can either: 1) export the page into a MAML file; or 2) export/preview an HTML version of their page converted using the MAML "translator", along with images zipped into the same output.

**Implementation.** We implemented the MAML editor as a web application that can run on any web browser. We used a microservices architecture, primarily consisting of a User Interface (UI) developed using Next.js and TypeScript, a Node.js API service to handle APIs and authentication, a MongoDB database for data storage, and a translator developed using Python and Selenium [6] for converting existing webpages into the MAML format.

Figure 2 shows a sample screenshot of the MAML editor user interface. On the back end, we implemented a web server, which is primarily responsible for managing APIs that facilitate several key functions: a) users' authentication, b) image(s) uploading, c) page

translation, and d) saving the pages, enabling users to continue their work from where they left off.

## 5.2 Translator

The MAML "translator" is designed to streamline the process of converting existing webpages into MAML, and vice-versa. The translator can transpile MAMLScript to JS, but not JS to MAMLScript, due to the complexity of JS, and the need of some human interaction to identify and trigger the set of functions needed. For this task, the translator is paired with the editor while controlled by a developer. An interesting avenue of future work is to explore the role of artificial intelligence in further automating the translator functionalities [8].

Given a webpage URL or local source code, the MAML translator initiates a Selenium Google Chrome instance and loads the specified webpage. To ensure that all resources, especially those with lazy loading, are fully loaded, the translator performs a sequential scroll through the entire webpage, and then returns to the top. Once the entire webpage has loaded, the translator conducts a Depth-First Search (DFS) of all HTML elements on the page and extracts the necessary information from them. For each supported element, the translator converts it into the corresponding MAML format while simultaneously recording the element's two-dimensional layout coordinates (x, y) on the page, dimensions (w, h) and its hierarchical positioning in terms of stacking order relative to other elements (z value). The x and y values are used for absolute positioning. The generated MAML file is then stored on the back-end server, and its corresponding public URL is returned in the API response, enabling it to be imported into the editor's user interface.

Although the MAML translator effectively handles many standard HTML elements, it currently struggles to accurately capture dynamic components, such as carousels and animated elements. These elements can update in real-time, but the translator only takes snapshots at specific moments, often missing transient states. Additionally, it does not fully represent animation and transition effects, as it might capture elements before their animations complete. Furthermore, the translator has limitations in capturing CSS properties defined at the parent level, such as content alignment and growth properties of child elements within a flex container. The MAML translator also encounters challenges with components that heavily rely on JS for rendering. These limitations indicate significant areas for further improvements.

## 6 MAML Benchmarking

This section benchmarks MAML with respect to: 1) user QoE measured with web performance metrics, 2) bandwidth savings, and 3) similarity with the originating webpage. We compare MAML webpages with both *original* and Google AMP [3] webpages, which was selected among the existing optimization tools since used by popular websites (e.g., `today.com` and `bbc.com`); further, it shares MAML's core principles (see Section 4.1). In the remainder of this section, we detail our methodology and present our analysis.

**Methodology.** We start by identifying a set of testing webpages. Given any webpage can be converted to MAML, we leverage the availability of AMP webpages as the key driver of our selection. Next, we generate MAML versions of these webpages using our translator. We do not include developers in this generation process so we can target a large(r) number of wepbages, while also benchmarking the performance of the sole MAML translator. We then rely on (student) developers, and a subset of webpages, to evaluate the full developing cycle envisioned for MAML.

We collect a list of AMP webpages using a methodology similar to [36]. First, we gather trending Google search queries from Google's *Year in Search* 2023 [29]. Next, we perform a Google search for each trending query and visit up to 100 webpages (up to page 10 of the search results) per query. For each webpage visited, we search for the `link` element with attribute `rel="amphtml"` located inside the head element. The `href` attribute of this element provides the AMP URL of a given webpage. We filter the list to include only one webpage per domain, resulting in a total of 115 webpages. From this list, we randomly select 100 properly working webpages.

Next, we use `webpagetest` [12] to automate the loading of these webpages in Google Chrome. Each webpage is loaded five times using network configurations representative of mobile networks in developing regions [43]: 1) **3G Slow**: 400 Kbps downlink/uplink rates with 400 ms RTT. 2) **3G Fast**: 1.6 Mbps downlink/768 Kbps uplink rates with 150 ms RTT. 3) **LTE**: 12 Mbps downlink/uplink rates with 70 ms RTT. A low-end mobile device (Xiaomi Redmi Go with a Quad-core 1.4 GHz CPU, and 1GB RAM) is used to access each version of the webpages 5 times.

As web performance metrics, we measure First Contentful Paint (FCP) [31], which is a user-centric metric measuring perceived load speed as it marks the first point in the page load timeline. Next, Speed Index (SI) [30] which measures how quickly a website's content is visually displayed during load. Finally, Page Load time (PLT) [41] which measures the amount of time it takes for a webpage to fully load. We also measure the total data (MB) consumed by each version of a webpage, and a visual similarity score obtained via a user study in Prolific [2].

**Results.** Figure 3(a) shows the Cumulative Distribution Function (CDF) of the delta size between a MAML and both an original (ORG) and AMP version of each of the 100 webpages under test, *i.e.,* a positive value indicates MAML data savings. Each value in the figure represents the median computed across 5 runs. The figure shows that MAML generates positive data savings for 90% of the webpages, with a median saving of 1 MB when compared to AMP and 2.4 MB when compared to original. Note also the long tail, with 50% of the savings spread between 1/2 MB and up to 15 MB; such large savings are possible since MAML webpages are rarely larger than 1 MB, as shown in the inset of figure 3(a). Note that about 10% of MAML webpages are larger than both original and AMP webpages due to the usage of image libraries which generate separate URLs for different image resolutions, updating based on the viewport size. However, the MAML translator could only capture the original, larger source file because different image libraries have separate ways to handle multiple source files, which the translator is not accustomed to, thus increasing the page size.

Next, Figure 3(b) shows boxplots of the (median) delta between the web performance metrics (FCP, SI, and PLT) measured for the original (ORG) and MAML version of each webpage under test, when considering variable network conditions (3G Slow, 3G Fast, and LTE). Accordingly, positive values represent MAML speedups;

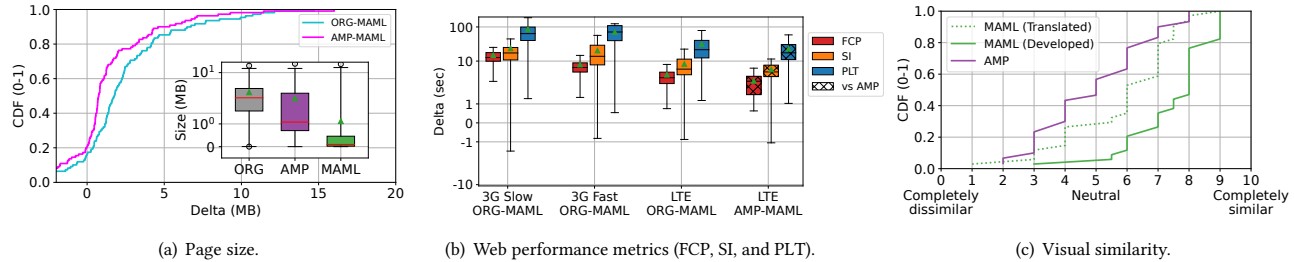

(a) Page size.

(b) Web performance metrics (FCP, SI, and PLT).

(c) Visual similarity.

**Figure 3: MAML vs. Original (ORG) and AMP in web performance metrics (FCP, SI, PLT), page sizes, and visual similarity.**

note that hatched boxplots refer to the delta of each metric when considering AMP as a baseline, and LTE, *i.e.,* the most challenging network condition for a potential speedup. Overall, the figure shows that MAML largely outperforms both original and AMP versions of each webpage, across all metrics and network conditions. As expected, the speedups are more prominent when considering worst network conditions, e.g., tens of seconds, regardless of the metric, when considering a "3G Slow" network. Still, even at LTE speed and when considering FCP, *i.e.,* the fastest metric, MAML shaves multiple seconds when compared to both original and AMP versions of the test webpages. The negative values are present because, for some pages, the MAML "translator" captured a higher resolution image, while these pages used image libraries to render a compressed image based on the viewport size.

Finally, we evaluate the similarity between MAML and AMP webpages with the originating webpage. To do so, we generate screenshots of each version of a fully loaded webpage and run a crowdsourcing campaign on Prolific [2] where we ask how similar each version of a webpage (AMP and MAML) is with respect to the original version of the webpage (see Appendix B for screenshots examples). We recruited 50 participants, each rating 10 screenshot pairs. Note that in this study Prolific testers cannot interact with the webpages, and can thus only evaluate their *visual similarity.* Please refer to the next section for a user study involving actual webpage interactions. We further limit this study to 50 webpages for which we also have MAML versions of the webpages generated by (student) developers (see the next subsection) which allows to evaluate the correctness of MAML translator.

Figure 3(c) shows the CDF of the median score received by each webpage version, with 0 indicating "completely dissimilar" and 10 indicating "completely similar". The figure shows that, for MAML, negative scores (0-3) are rare (about 10% of the scores) even when webpages are "translated", *i.e.,* only generated by the translator with no human intervention. Still, the role of a developer is not negligible to achieve high visual similarity score, with an overall score improvement of two points, on average. Last but not least, MAML outperforms AMP by one point when translated and up to 3 points when allowing a developer in the generation process.

## 7 MAML Usability

In this section, we introduce developers in the process of creating MAML webpages. Our main goal is to evaluate the *usability* – in terms of user interaction – of MAML webpages. In addition, we

aim to validate both the *speedup* and *data savings* of the translator, when considering "fully" functional MAML webpages.

**Methodology.** We recruited 25 students from an international university to participate in a competition to create MAML webpages which closely resemble their original versions, both in term of visual aspect and interactivity. The competition offered prizes for the first (iPhone 13), second (iPad), and third-place (AirPods) winners. The competition was conducted asynchronously, *i.e.,* students were allowed to work on creating MAML webpages on their own over the course of two weeks. Each student was given 2 unique webpages randomly extracted from the 100 webpages from Section 6. The students were given an introduction on how to use the MAML editor; further, an institutional review board (IRB) approval was granted to conduct the user study, and the authors who conducted the study are CITI [7] certified. No sensitive or personal information of the participants is collected, except for their university email address required to contact them for the prize.

Before the competition, participants filled out a form asking about their expertise in web development and how important page load time is for them in building a website. With regards to experience, out of all participants, 2 had no web experience, 6 were beginners, 7 were intermediate developers, and 5 had advanced web experience. As for the importance of page load time, the majority of participants responded with either 4 or 5 (5 indicating "extremely important", and 0 indicating "not at all important"). Appendix A describes the details of the survey.

**Results.** We start by extending the results from Figure 3(c) when considering 50 MAML webpages produced by our student developers and judged by five *expert* evaluators. Differently from before, we now ask the evaluators to interact with a MAML webpage and *stress test* it, *i.e.,* explore all its functionalities to the best of their ability. For this reason, we discard Prolific and resort to five expert developers who are tasked to evaluate: 1) the functional similarity between each MAML and original page, 2) the impact of eventually missing functionalities. Both questions are answered on the usual scale comprised between 0 ("completely dissimilar" and "no impact") and 10 ("completely similar" and "extreme impact"). Figure 4(a) summarizes the responses collected for both questions. The figure shows that all webpages have very high scores (6-10) with respect to functional similarity (green bars), indicating that most webpages generated closely mimic the original webpages. With respect to the impact of the missing functionalities (red bars), most scores are comprised between 0 and 4, suggesting either no impact

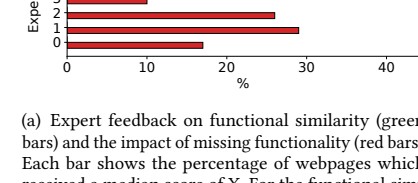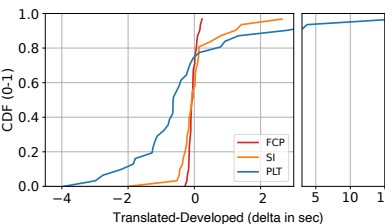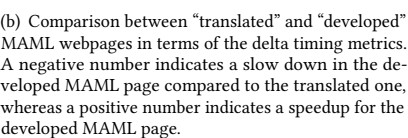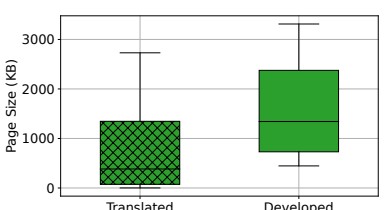

(a) Expert feedback on functional similarity (green bars) and the impact of missing functionality (red bars). Each bar shows the percentage of webpages which received a median score of X. For the functional similarity (green bars) the score goes from 0 (completely dissimilar) to 10 (completely similar), whereas for the impact of missing functionality (red bars) the score goes from 0 (no impact) to 10 (extreme impact).

(b) Comparison between "translated" and "developed" MAML webpages in terms of the delta timing metrics. A negative number indicates a slow down in the developed MAML page compared to the translated one, whereas a positive number indicates a speedup for the developed MAML page.

(c) Size analysis of "translated" and "developed" MAML webpages. The hatched boxplot represents the "translated" MAML pages, whereas the solid boxplot represents the "developed" MAML pages.

**Figure 4: Developed MAML webpages in terms of their functional similarity to Original webpages, their size comparison to the "translated" MAML webpages, and the delta size ("translated" - "developed") correlation as a function of webpage complexity.**

or moderate impact. While empirically evaluating the missing functions, we found that these functions are currently not supported by MAMLScript (see Section 4.3), e.g., interactive graphs and user triggered animations, but can be supported in the future.

We evaluate the inter-rater reliability of the expert annotations using the Intraclass Correlation Coefficient (ICC)—a statistical measure used to assess the reliability or consistency of measurements made by different raters or across repeated measurements of the same subject. The ICC values are shown in Table 5, and are derived using the ICC(3,k) model, which is appropriate for a fixed set of raters providing ratings on a common set of images [37]. The Intraclass correlation value was 0.734, indicating good inter-rater reliability with a statistically significant agreement, with p-values well below 0.05, further validating the consistency of the ratings.

Next, we set out to validate the *speedup* and *data savings* obtained by the translator in Section 6. Figure 4(b) shows the CDF of the delta between the web performance metrics (FCP, SI, and PLT) measured for the "translated" and "developed" 100 webpages, *i.e.,* a negative value indicates a *slowdown*, when considering LTE (*i.e.,* the most challenging network condition for a potential speedup). Overall, the figure shows a slowdown for ∼78% of the webpages due to the extra content added by the developers while "fixing" the translated webpages (1 MB at the median as shown in Figure 4(c)). This had almost a negligible effect on the fast FCP (less than 280ms), while added 1/2 seconds for 5% of the webpages in term of SI (and less than 500 ms for the remainder 73% of webpages). PLT is the most affected metric, for which 20% of the webpages had a slowdown of roughly 1.5 - 4 seconds. This is expected as PLT measures the amount of time it takes for a webpage to fully load, and it is thus impacted the most by the larger size. Even with these corrections, MAML

still offers considerable webpages speedups over both original and AMP webpages (see Figure 3(b)).

Finally, Figure 4(b) shows show considerable speedups – up to 15 seconds for PLT – for about 22% of the developer webpages. These speedups are due to image optimizations done by our developers, who have properly selected lower resolution images compared to higher resolutions picked by the translator, as previously discussed.

## 8 Conclusion

This paper has presented the Mobile Application Markup Language (MAML), a flat layout-based web specification language that reduces computational and data transmission demands, thereby accelerating and slimming webpages to improve the web quality of experience of users in developing regions. To demonstrate and evaluate MAML, we have developed a web-based *editor* and recruited 25 students to compete in porting popular webpages to MAML. We have further developed a *translator* which allows to automate this conversion, while missing complex page functionalities related to web page interaction. We use the translator to benchmark 100 popular webpages which also support AMP, a Google format which rewrites webpages with new HTML tags and elements optimized for performance. Our analysis shows that MAML vastly outperforms AMP, accelerating webpages by tens of seconds under challenging network conditions thanks to its very compressed format (50-80% page size reduction). Further, these performance optimizations are achieved while generating webpages which adhere more to the original webpages than what AMP can achieve. With respect to page functionalities, a user study shows that MAML is quite effective in maintaining the most important functionalities when pairing the translator with some developers help.

| | ICC | 95% CI | | F Test with True Value 0 | | | |
|---|---|---|---|---|---|---|---|
| | | Lower Bound | Upper Bound | Value | df1 | df2 | P value |
| ICC(3,k) | 0.734 | 0.56 | 0.85 | 3.76 | 33 | 132 | $3.2 \times 10^{-8}$ |

**Table 5: Inter-rater reliability of our expert evaluation on the functional similarity results.**

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

## A Survey Questionnaire and Results

| Question | Options |
|---|---|
| How much web development experience do you have? | A. None |
| | B. Beginner (understand the basics, can use templates and customize them) |
| | C. Intermediate (can develop pages from scratch and write limited JS code for interactivity) |
| | D. Advanced (have developed webpages from scratch using modern web development technologies and can write JS code from scratch) |
| How important is page load time for you when developing webpages? Rate on a scale from 0 to 5. | 0 - Not at all important |
| | 5 - Extremely Important |

**Table 6: Pre-competition survey questionnaire**

| Question | Options |
|---|---|
| How would you rate the learning curve of the MAML Editor on a scale from 0 to 10? | 0 - Extremely Hard |
| | 10 - Very easy to learn |
| Rate MAML Editor's web interface on a scale from 0 to 10. | 0 - Terrible |
| | 10 - Excellent |
| Rate the MAML editor usability on a scale from 0 to 10. | 0 - Unusable |
| | 10 - Easy to use |

**Table 7: Post-competition survey questionnaire**

| Question | Options |
|---|---|
| Rate the visual similarity of the two pages on a scale from 0 to 10. | 0 - Not similar at all |
| | 5 - Moderately similar |
| | 10 - Identical |
| Rate the visual impact of the missing content on the user experience on a scale from 0 to 10. | 0 - No impact |
| | 5 - Moderate impact |
| | 10 - Extreme impact |
| Rate your willingness to sacrifice missing content for a significant increase in loading speed. | 0 - Not willing at all |
| | 5 - Moderately willing |
| | 10 - Extremely willing |

**Table 8: Content similarity study questionnaire on Prolific**

| Question | Options |
|---|---|
| Rate the functional similarity of the two pages on a scale from 0 to 10. | 0 - Not similar at all |
| | 5 - Moderately similar |
| | 10 - Identical |
| Rate the functional impact of the missing content on the user experience on a scale from 0 to 10. | 0 - No impact |
| | 5 - Moderate impact |
| | 10 - Extreme impact |

**Table 9: Functional similarity study questionnaire for manual inspection**

# B Sample Original vs. MAML pages

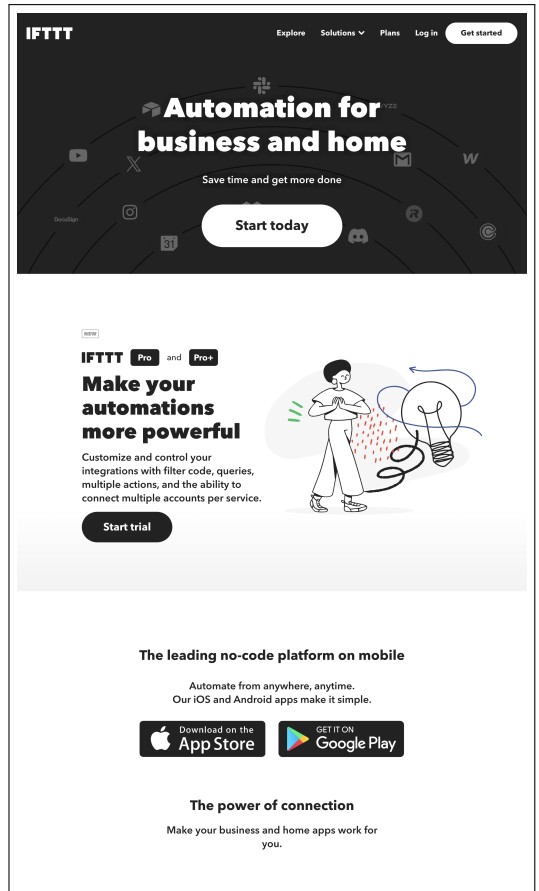

(a) ifttt.com original page

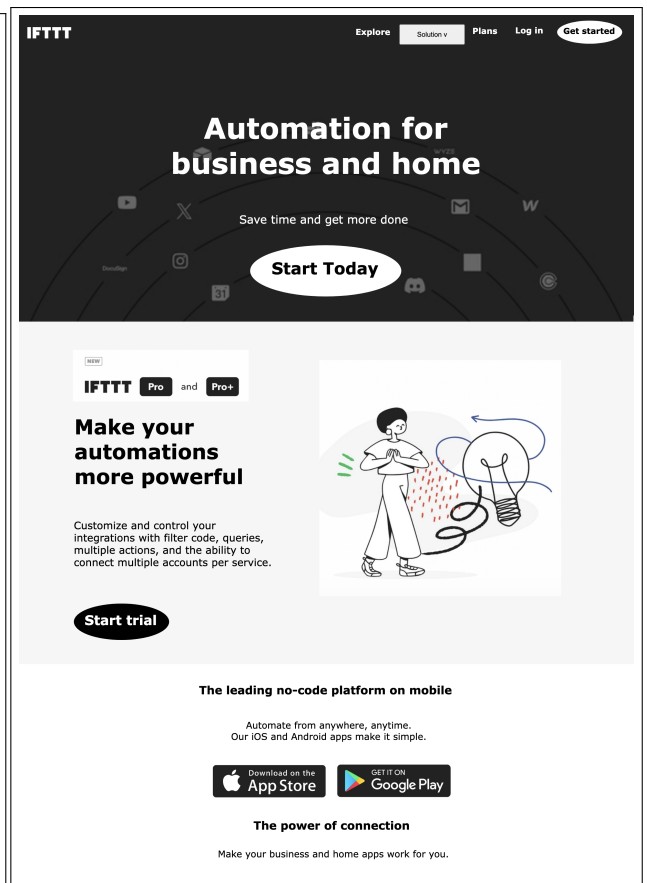

(b) ifttt.com MAML page

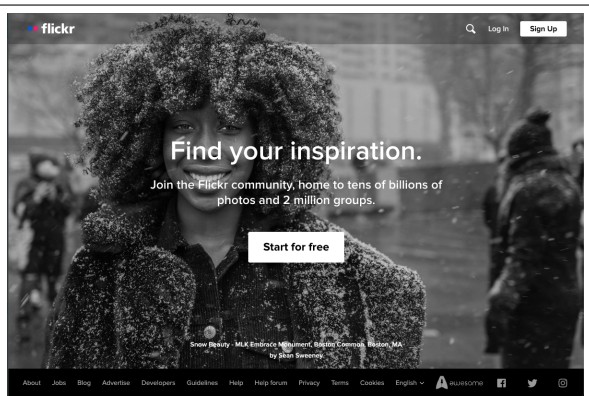

(c) flickr.com original page

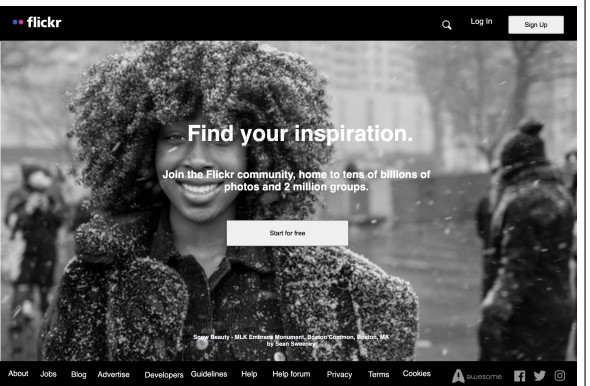

(d) flickr.com MAML page

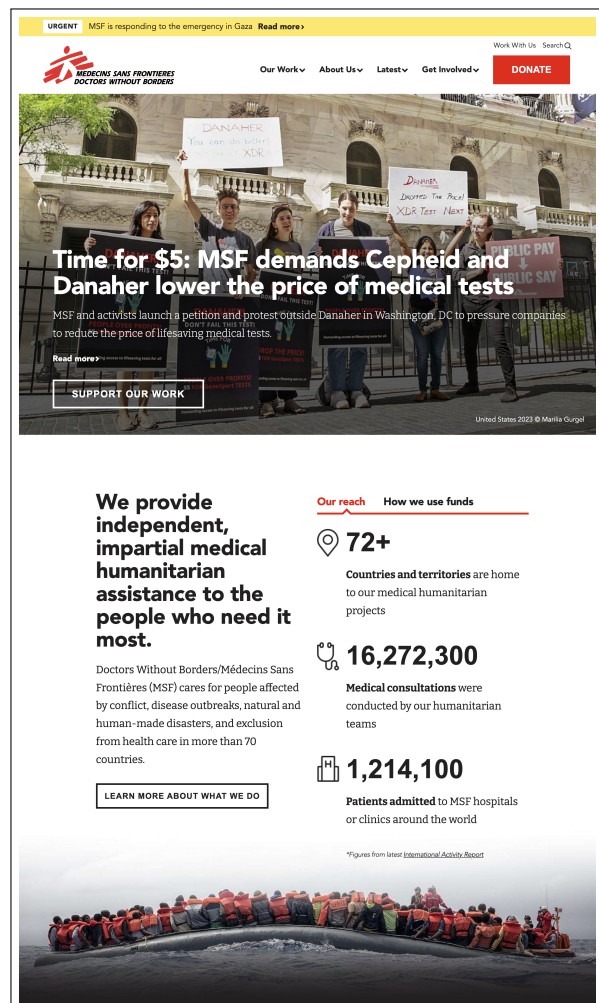

(e) doctorswithoutborders.org original page

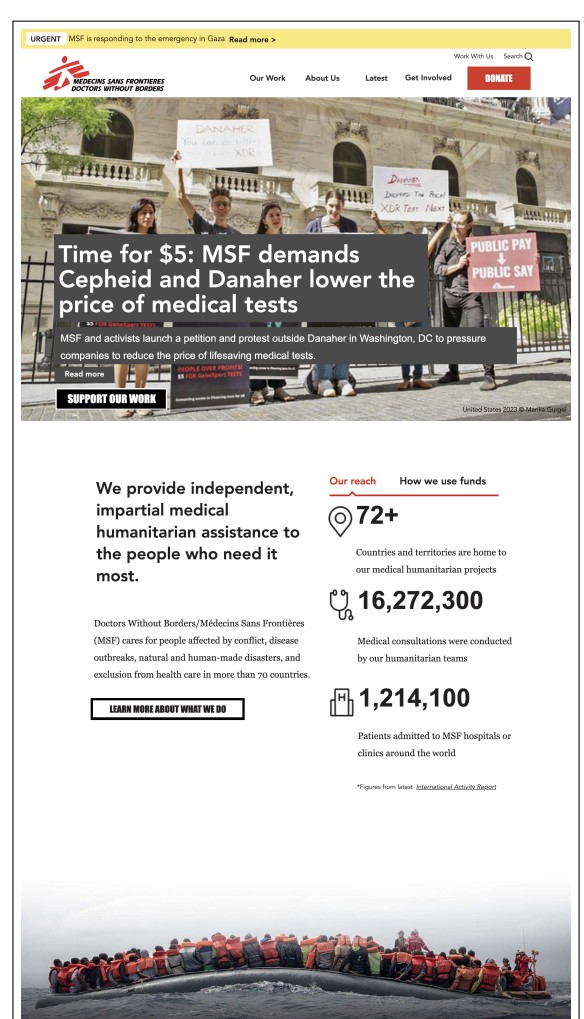

(f) doctorswithoutborders.org MAML page

