# OpenReview forum: "MAML: Towards a Faster Web in Developing Regions"
_ACM.org/TheWebConf/2025/Conference — WWW 2025 Oral_

### Official Review · Reviewer_RZu8 · 2024-11-25

**Novelty:** 6
**Technical Quality:** 3

**Review:**

This paper introduces MAML, an alternative to the HTML+CSS+JS stack, aiming to significantly reduce the size of Web applications and improve their runtime performance by trimming the functionality of HTML+CSS+JS into a strict subset. The paper first highlights the increasing bloat in Web applications based on the current technology stack, emphasizing the challenges this poses, especially in regions with limited network resources and underpowered devices. Then, it elaborates on the three core designs of MAML, which include:

- Abandoning all other layout mechanisms and adopting a flat DOM based entirely on absolute positioning.
- Trimming JavaScript to retain only a minimal subset of core APIs, referred to as MAMLScript.
- Providing an automated translator for converting MAML to HTML+CSS+JS, along with a visual editor to facilitate application migration.

The core idea of MAML bears resemblance to Google’s AMP technology, and the experimental section of the paper compares MAML with AMP. Finally, the experiments demonstrate that applications developed using MAML outperform their HTML+CSS+JS and AMP counterparts in terms of size and load speed, while maintaining similarity to the original HTML+CSS+JS version (which can be seen straightforward in the screen snapshots of Appendix B).

## Pros

1. The paper identifies the issue of bloated Web applications and proposes a solution to address it. Given the diverse scenarios in which Web applications are used, this technology is not only relevant for developing regions but also for environments in developed regions with limited networks or resource-constrained edge or embedded devices. This adds to the universality of the proposed approach.

2. The proposed idea is highly innovative and bold, representing an admirable attempt to rethink Web application design.

3. The work presented in the paper is rigorous. The design of MAML, its translator, and editor appear to have required significant effort and have achieved a high degree of completion and practical utility.

4. The experiments are comprehensive, evaluating both the performance and usability of MAML, and claim to deliver results superior to Google AMP.

## Cons

1. The design of MAML seems overly aggressive, even more so than Google AMP, which has itself faced criticism. This could greatly hinder its adoption and acceptance.

2. The modern Web front end has evolved far beyond static information pages, now serving as client-side applications with complex logic and functionality. In service systems adopting a front-end-backend separation architecture with Web APIs (e.g., REST), the server merely transmits abstract data, and the rendering is entirely handled by front-end JavaScript. It is unclear whether MAMLScript, a heavily restricted subset of JavaScript, can adequately support such applications.

3. The paper does not explain how MAML pages (which appear to be merely a sequence of JSON objects) are rendered. Did the authors modify Chrome's source code to support MAML rendering? Or do MAML pages require client-side browsers to preload a JavaScript bootstrap script to generate the corresponding DOM elements? If the latter is true, doesn’t this imply that MAML is still based on HTML+JS and could essentially be regarded as a Web page compression technique?

**Questions:**

1. How does MAML support Web applications that use Web APIs and are developed in a front-end-backend separation architecture, where the front end needs to decode the responses and dynamically generate DOM elements?

2. Is MAML still based on HTML+JS? Or does it require a specialized browser or mobile app to access? If so, how can it achieve better performance than HTML+JS in practice?

3. MAML relies entirely on absolute positioning for layout and scales the entire interface to adapt to different screen sizes. However, in real-world applications, the scaling ratios of the elements in the page are not the same. For example, fixed-size fonts may become too small to read after scaling. How does MAML handle this issue?

**Reviewer Confidence:**

3: The reviewer is confident but not certain that the evaluation is correct

**Scope:**

4: The work is relevant to the Web and to the track, and is of broad interest to the community

---

### Official Review · Reviewer_XXYd · 2024-11-29

**Novelty:** 5
**Technical Quality:** 2

**Review:**

I would like to thank the authors for the paper. The paper presents an interesting solution to allow the possibility of accessing web pages under unfavourable conditions (poor network connectivity and low-end devices with limited capabilities).

I appreciate the technical contribution of the paper, i.e., the MAML web specification language. I think it may be useful in some use cases. However, I think the paper needs some further work before it is ready to be published in a top venue like WWW. If WWW had a one-shot review option, I'd go with that one, since most of my comments can be addressed in few months. Unfortunately, WWW does not offer this option and I don't think most of the issues I raised are fixable in a shepherding process.

Overall my main concerns are the followings:

1- The authors do not provide convincing arguments to support in which use cases a framework like MAML would be useful. They try to make the case for the use case in developing countries, but the fail to support this use case in a convincing manner.
In addition, authors disregard the implications of the large simplification imposed by MAML in comparison with current webs. In particular, the Web is funded to a great extend by online advertising. Frameworks like MAML may simply kill online advertising in its current form of operation, what would eliminate the incentive for supporting the most important webpages in developing countries. This is an important discussion to have.


2- The evaluation needs to be largely improved. On the one hand, the objective evaluation (i.e., experiments to measure web performance metrics) is conducted in likely unrealistic setup based on likewise unrealistic assumptions. The considered scenario does not seem to be representative of developing countries. On the other hand, the conducted subjective evaluation is not rigorous enough and is not convincing regarding the actual user experience expected from MAML vs other considered benchmark.




I provide extensive and detailed comments below that I hope authors find useful. I encourage authors to follow up their work since a more convincing framing and usecases and an improved evaluation are likely to make this a strong paper.



Section 2

- References are quite old. All of the references used in the Challenging Network Conditions are around 10 years all. For instance Ref 53 to justify large loading times is from 2014 (10 years ago). Ref 38 from 11 years ago.

I recommend the authors to have a look to updated data, e.g. https://www.speedtest.net/global-index.
To name a country authors use in the paper, India presents avg connection speeds of 95Mbps in mobile devices and 63Mbps in desktop devices, based on the speedtest dataset.

Authors should convince the reader to what extend the highlighted issue is a real problem nowadays, in which countries and to how many people it affects. This is fundamental to understand the actual need/impact of the proposed solution.


- References in the Web Complexity are again quite old (around 10 years). Does the findings in those papers still hold today?. Webs might be way more complex now than what reported in those papers, or contrary they may have became more efficient.


- Again in the paragraph "Optimization" authors refer to papers (Refs 11, 42, 50) from a decade ago as "recent works".



- Authors do not discuss the penetration and use of existing solutions trying to address the same problem such as Google AMP, Facebook Instant Articles  or SpeedReader. If this was a widespread problem these solutions would be largely used. Is this the case?.


Section 3

The motivation is based on artificial setup handpicked by the users, which we do not know if is representative of developing countries.
One one hand, the authors do not describe the characteristic of the emulated low-end device. And they do not actually provide data on the type of devices used in different developed countries. On the other end the authors set up some artificial network conditions, which are not justified in any data related to current network conditions from developed countries.

Finally, they do not indicate from which location they perform the experiments. This might have potential implications. If I run an experiment from a lab in US requesting local websites from Pakistan or Ghana, it is likely that the web is not cached locally by CDNs and the loading time requires larger loading times.


Section 4

4.1. Design principles

Many websites, and specifically those attracting a larger fraction of traffic (e.g., newspapers or e-commerce sites) are funded mainly or partially based on online advertising. Online advertising is responsible for a large fraction of the JS code present in a website.
The proposal by the authors implies as they state "a completely cutting off JS".
This may seriously impact the business model of many websites in the current World Wide Web.

Making more general this comment. The authors should discuss and study the practical implications of their proposed solution for the business model funding the current Web.


4.2 Supported elements

The supported elements leave aside some ad formats such us interstitial and others. Which other elements from current webs are left aside?.
This is again important to be reported by authors to understand the limitations of their proposed solution.


4.3 MAML Script

Supported functionality -> Authors use an outdated reference such as Alexa, which is a discontinued project by Amazon.
Moreover they only consider 100 webs to manually identify the top used JS functions.


5.2 The MAML translator

Authors claim clean about the limitations, acknowledge them.
However they are not clear about how these limitations affect translation of webs at the current moment. Is it a ready to use tool? How many webs would be affected by these limitations (for instance using the same 100 than in Section 4.3).


Section 6. Benchmarking

I have many concerns about the experiment set up, which makes me to have similar concerns about the representativeness of the obtained results. Authors do not provide convincing arguments to support their methodology to select websites, network conditions and device for their experiments. The assumptions made (specially in the case of the network conditions and mobile devices) are likely representing worse conditions compared to the reality. This would be in the benefit of the proposed solution, which due to its nature will offer better performance than other alternatives when the set up is worse (i.e., worse network conditions and device capacity).

Instead of making such weakly supported assumptions I would recommend authors to reframe this section and to avoid selecting a specific set up. Instead authors could reproduce the experiments for a range of scenarios in terms of network conditions and device capacity. Then, they can make an exhaustive analysis in which they can identify the range of conditions in which their proposed solution outperforms the alternative considered solutions. This is a much more meaningful and useful analysis for web developers, which can decide when MAML might be an option and when not.

Further details can be found below:


It is not very clear how the selection process of webpages has been done. If I understood it correctly, they find the categories of Google year in trends (I checked and they offer 5 trends in more than 15 categories). This means +75 categories. For each of them, the authors collect 100 webpages. This makes a total of 7500 webpages.

From these they identify which ones have the AMP URL (i.e., implement AMP). The authors claim it is just 115 from which the authors claim end up selecting 100 randomly "which work properly".

I've several concerns from this process:

1- If we make the statistics about these dataset only 115 out of +7500 implement AMP, i.e., are interested in solutions which makes their web more accessible (~1,5%). This indicates a low interest from developers in implementing these type of solutions.
This opens the question about the actual need of these solutions?.

2- Why the selection is done in this way?. If the focus are developing countries, the webs selected should have been webs popular in these countries. Maybe AMP is more popular in these webs.

3- The dataset is very limited (just 100 webs) and biased from the reasons in the previous points. Why limiting the comparison to those webs implementing AMP?. A much larger number of webs should have been considered in the comparison with original webs.



I've also serious concerns regarding the selection of the network configurations in developing countries. Based on the Speedtest report (https://www.speedtest.net/global-index) just 2 countries of the 111 considered have an average download speed in mobile networks below 12Mbps, which is the best case scenario presented by authors. For web pages the important side of connection is downstream, since the upstream is not that relevant (the device sends small queries of few Kbytes to the web server)
Hence, the selection of network configurations seems to be unrealistic, and thus the results are likewise unrealistic. The considered scenario are biased in such a way that the proposed solution shows better performance since it clearly significantly reduce the number of bytes being transferred.

Finally I've also have concerns about the selection of the mobile device. Why is this representative of developing countries?. Authors should provide convincing



Section 6. Results

As explained above I have concerns about the reported performance of MAML vs other considered alternatives (original and AMP) since the considered scenario is likely unrealistic.

Second, it is unclear the decision taken by the authors regarding subjective evaluation. The main problems that can derive from the use of MAML is due to the interactivity of the web since it reduce a lot the use of JS. Hence, the validation should expose the users to an actual interaction with the webpage and then obtain their evaluation regarding the user experience of original vs AMP vs MAML.

Despite authors have a group of users from prolific selected to run subjective evaluation, they decided to not use these and instead selected a group of 5 experts. However, no description about who the 5 experts were? Were they independent experts? Do they have any relation with the paper, the authors or the institution of authors?.

The argument from authors to drop prolific users from this evaluation is very weak (i.e., they are not experts). However, the typical user of webpages is not an expert developer and thus it is paramount to validate the user experience with this group of users. So the authors should have run the experiment with Prolific users and then compare the results obtained from the experts vs. prolific panel.

In addition, a description of the prolific panel must also be added. Which is the demographic of the panel? (Age and genders).

In summary, the subjective evaluation needs to be improved to be credible.

**Questions:**

1. Can you provide solid and more recent references that justify why developing countries may need solutions like MAML? Why if this is so, solutions like AMP are not widely used?

2. Can you discussed the implications of using MAML in online advertising, which is the main source of funding of the web?

3. Can you provide solid arguments to justify the selected set up for evaluation is realistic?.

4. Can you provide further details about the 5 experts you used in your evaluation?

5. Can you justify why you disregard prolific users from the evaluation of user experience?

6. Please consider the rest of the comments in the review above

**Ethics Review Description:**

No needed

**Reviewer Confidence:**

3: The reviewer is confident but not certain that the evaluation is correct

**Scope:**

4: The work is relevant to the Web and to the track, and is of broad interest to the community

---

### Official Review · Reviewer_1njZ · 2024-11-30

**Novelty:** 6
**Technical Quality:** 6

**Review:**

### Summary
Nowadays, webpages are bloating; thus, loading them requires more time than before. Moreover, their functionalities are becoming complex. Since some users are still using devices with insufficient computing power and are not connected to a fast enough network, loading these web pages will consume a lot of time, greatly affecting the user experience.

This paper presents a new language, MAML (Mobile Application Markup Language), which can describe a webpage. It follows the following design principles, i.e., flat DOM, MAMLScript to mimic JS, and backward compatibility. The comprehensive evaluation results illustrate its practicability.

### Strength
- Strongly supported motivation
- Comprehensive evaluation design

### Weakness
- Not open-source yet

### Comments
To be honest, I enjoy reading this paper. The paper is well-organized, and most of the statements are strongly supported by experiment results or other materials. But I did not notice that the paper gave a link to the MAML editor, whose release, I think, will benefit the whole community.

Moreover, here are some comments and issues that should be discussed or addressed.

Since MAML uses proportional scaling to handle element positions, will it be very difficult to click on certain elements on some low-resolution or small-screen devices? Whether this method affects the user experience should also be an indicator in the user study.

In the MAMLScript part, the paper mentioned that some JS functions, like scroll-triggered animations, are not yet supported. Please discuss the difficulty and feasibility of adding support for them.

Some marks are not well introduced in Figure 3(a) and 3(b), like the triangle and the red horizontal line.

In Section 7, I would like to know how much effort these students take to translate a webpage.

**Questions:**

Please refer to the last four paragraphs in the Review part.

**Reviewer Confidence:**

3: The reviewer is confident but not certain that the evaluation is correct

**Scope:**

4: The work is relevant to the Web and to the track, and is of broad interest to the community

---

### Official Review · Reviewer_xr1S · 2024-12-03

**Novelty:** 5
**Technical Quality:** 6

**Review:**

-addresses an important real-world problem
-professionally written and reported, including figures, table, and equations
-evaluation study showing performance improvement

What could be improved is the next steps toward broader implementation and adoption---what are the next steps for driving positive change with MAML?

**Questions:**

What are the next steps for driving positive change with MAML?

**Reviewer Confidence:**

2: The reviewer is willing to defend the evaluation, but it is likely that the reviewer did not understand parts of the paper

**Scope:**

4: The work is relevant to the Web and to the track, and is of broad interest to the community